# Role of Inflammatory Mechanisms in Major Depressive Disorder: From Etiology to Potential Pharmacological Targets

**DOI:** 10.3390/cells13050423

**Published:** 2024-02-28

**Authors:** Bruna R. Kouba, Laura de Araujo Borba, Pedro Borges de Souza, Joana Gil-Mohapel, Ana Lúcia S. Rodrigues

**Affiliations:** 1Department of Biochemistry, Center of Biological Sciences, Universidade Federal de Santa Catarina, Florianópolis 88040-900, SC, Brazil; bruna.kouba@posgrad.ufsc.br (B.R.K.); laura.borba@posgrad.ufsc.br (L.d.A.B.); pedro.b.souza@posgrad.ufsc.br (P.B.d.S.); 2Island Medical Program, Faculty of Medicine, University of British Columbia, Victoria, BC V8P 5C2, Canada; 3Division of Medical Sciences, University of Victoria, Victoria, BC V8P 5C2, Canada

**Keywords:** anti-inflammatory approaches, gut dysbiosis, inflammation, major depressive disorder

## Abstract

The involvement of central and peripheral inflammation in the pathogenesis and prognosis of major depressive disorder (MDD) has been demonstrated. The increase of pro-inflammatory cytokines (interleukin (IL)-1β, IL-6, IL-18, and TNF-α) in individuals with depression may elicit neuroinflammatory processes and peripheral inflammation, mechanisms that, in turn, can contribute to gut microbiota dysbiosis. Together, neuroinflammation and gut dysbiosis induce alterations in tryptophan metabolism, culminating in decreased serotonin synthesis, impairments in neuroplasticity-related mechanisms, and glutamate-mediated excitotoxicity. This review aims to highlight the inflammatory mechanisms (neuroinflammation, peripheral inflammation, and gut dysbiosis) involved in the pathophysiology of MDD and to explore novel anti-inflammatory therapeutic approaches for this psychiatric disturbance. Several lines of evidence have indicated that in addition to antidepressants, physical exercise, probiotics, and nutraceuticals (agmatine, ascorbic acid, and vitamin D) possess anti-inflammatory effects that may contribute to their antidepressant properties. Further studies are necessary to explore the therapeutic benefits of these alternative therapies for MDD.

## 1. Introduction

Major depressive disorder (MDD) is characterized by depressed mood and/or loss of interest or pleasure, together with changes in appetite and sleep, pessimism associated with feelings of guilt and/or worthlessness, impaired concentration, decreased energy or fatigue, along with other symptoms [1]. It is one of the most frequent and disabling psychiatric disorders, affecting all aspects of life. Estimates indicate that 3.8% of the population suffers from MDD worldwide, with a prevalence of 5% in adults and 5.7% in the elderly, being 50% more common in women than in men. In severe cases, it can lead to suicide, which is the cause of death of more than 700,000 people every year [2].

Although the pathophysiology of MDD is not yet fully elucidated, this complex and multifactorial disorder is thought to involve multiple genetic, environmental, and neurobiological factors [3]. Indeed, approximately 50% of patients present with MDD that is refractory to classic antidepressant drugs, thus disputing the monoaminergic hypothesis of depression and suggesting that other monoamine-independent mechanisms are also involved in the neurobiology of MDD [4,5].

Several lines of evidence have pointed towards a strong relationship between inflammatory processes and the pathophysiology of MDD. Peripheral proinflammatory cytokines can reach the brain and activate glial cells, leading to neuroinflammation and affecting behavior and emotions [6,7]. Increased proinflammatory cytokines are also associated with other mechanisms involved in MDD, including hyperactivity of the hypothalamic-pituitary-adrenal (HPA) axis, dysfunction of the glutamatergic system, impairment of neuroplasticity, dysbiosis of the gut microbiota, and alterations in tryptophan (TRP) metabolism [3,8,9].

Indeed, evidence has shown that systemic inflammatory markers, mainly interleukin (IL)-6, tumor necrosis factor-α (TNF-α), and C reactive protein (CRP) are commonly elevated in MDD-affected individuals as compared to controls, suggesting the presence of chronic low-grade inflammation in MDD [10,11,12]. Importantly, immune system activation is observed in both clinical studies and in animal models of MDD, since the relationship between the peripheral immune system and neuroimmunological mechanisms is associated with the onset and maintenance of depressive symptoms [6,7]. In particular, animal studies have demonstrated that peripheral cytokines can affect brain circuits, behaviors, and mood directly and indirectly [13,14]. Moreover, patients treated with cytokines such as IL-2 and interferon-γ (IFN-γ) for viral infections or cancer may develop depressive symptoms [15]. Additionally, individuals with autoimmune and chronic inflammatory diseases, including rheumatoid arthritis, multiple sclerosis, fibromyalgia, and inflammatory bowel disease are likely to experience depressive symptoms [16,17]. On the other hand, managing exacerbated inflammation can alleviate depressive symptoms and slow down the progression of both MDD and its comorbidities [18,19,20].

Several clinical and preclinical studies have investigated how the immune system and inflammatory pathways are involved in the neurobiology of MDD and have suggested that MDD can be viewed as a systemic disorder [6,7]. In line with this idea, the modulation of inflammatory mechanisms and promotion of homeostasis have been associated with the beneficial effects of traditional antidepressants. This review aims to highlight the main inflammatory mechanisms involved in the pathophysiology of MDD, including the roles of glial cells, the peripheral immune system, and gut dysbiosis. Additionally, new therapeutic approaches for this psychiatric disorder, including physical exercise, probiotics, and some nutraceuticals that have anti-inflammatory properties will also be discussed. Such alternative therapeutic strategies are particularly promising, considering the limitations of the currently available antidepressants and the fact that poor responses to classic antidepressants have been associated with aberrant inflammatory processes [21].

## 2. Neurobiology of Major Depressive Disorder

Stress is a well-characterized environmental risk factor for MDD, leading to the activation of the HPA axis [22,23]. This results in the secretion of glucocorticoids by the adrenal cortex, which can then activate glucocorticoid receptors (GRs) expressed throughout the body. In turn, glucocorticoids can elicit several genomic and non-genomic physiological processes that affect several metabolic, immunological, and cognitive functions [24].

Several clinical and pre-clinical studies have demonstrated that HPA axis abnormalities, including glucocorticoid hypersecretion, GR resistance, and loss of negative feedback, are found in MDD [25]. In fact, a high percentage of MDD individuals have higher plasma and salivary cortisol levels when compared with healthy controls [26]. Furthermore, a meta-analysis showed a predictive effect of cortisol levels on the onset of MDD [27], while Cattaneo et al. [28] demonstrated that treatment-resistant and non-treated MDD patients had glucocorticoid resistance along with increased levels of pro-inflammatory cytokines, as demonstrated using whole-blood mRNA expression analysis. In addition, using binomial logistics models, a signature of GR, P2X purinoceptor 7 (P2RX7), IL-1β, IL-6, TNF-α, and CXC motif chemokine ligand 12 (CXCL12) mRNAs was shown to discriminate treatment-resistant patients from responsive patients.

In pre-clinical studies, stress exposure or chronic administration of corticosterone (analog of cortisol) in rodents is known to induce depressive-like behaviors, whereas pharmacological inhibition of corticosterone synthesis can prevent stress-induced depressive-like behaviors, further emphasizing the involvement of the HPA axis in MDD [29,30]. A dysfunctional HPA axis is associated with an abnormal pro-inflammatory cytokine profile [31]. In addition, chronic mild stress was shown to result in depressive-like behaviors, along with an increase in the brain levels of pro-inflammatory cytokines (IL-1β, TNF-α, and IL-6), and a decrease in the brain levels of anti-inflammatory cytokines (transforming growth factor-β (TGF-β) and IL-10 in the brain) [32,33,34]. Furthermore, IL-6 has been associated with increased activity of the HPA axis by increasing cortisol levels and coordinating biological pathways underlying stress and stress-induced depression [35].

Another mechanism that has been widely investigated is the dysfunction of the glutamatergic system and its relationship with impairments in neurogenesis and synaptogenesis, all of which have been implicated in MDD [36]. The involvement of the glutamatergic system in MDD is supported by the rapid antidepressant effect of ketamine, an N-methyl-D-aspartate (NMDA) receptor antagonist [37]. Mechanistically, the blockage of NMDA receptors by ketamine favors the activation of α-amino-3-hydroxy-5-methyl-4-isoxazolepropionic acid (AMPA) receptors in pyramidal neurons, culminating in a calcium influx through voltage-dependent calcium channel with the consequent release of the brain-derived neurotrophic factor (BDNF), which in turn activates pathways related to neurogenesis and synaptogenesis, thus improving depressive symptoms [38]. Research has shown that immuno-inflammatory pathways, including neuroinflammation, with microglial activation, astrocyte atrophy, and the release of inflammatory cytokines, are involved in the dysfunction of the glutamatergic system found in MDD [7,39,40].

Peripheral inflammatory pathways can also reach the central nervous system through several mechanisms, including the gut-microbiota-brain axis, which plays an important role in psychiatric disorders, including MDD [41]. Recent studies have evaluated the correlation between altered gut microbiota and MDD. Indeed, it is often hypothesized that normalizing gut microbiota can improve depressive symptoms [42]. Importantly, inflammation resulting from alterations of the gut-microbiota-brain axis has a significant adverse impact on neurotrophin levels, which are critical in overcoming depressive symptoms by maintaining synaptic plasticity [43]. Also, the dysregulation of the HPA axis found in MDD is commonly influenced by neuroinflammation, which is often caused by an imbalance in the gut-microbiota-brain axis [44]. Importantly, many studies have also demonstrated that changes in the gut microbiota trigger inflammatory cytokines, such as IL-6, IL-1β, IL-2, and IFN-γ. These cytokines can reach the brain through neuroanatomical and neuroendocrine pathways, thus affecting mental health and behavior [14,15,16,17,45].

Tryptophan metabolism and the kynurenine (KYN) pathway (which can be influenced by increases in inflammatory cytokines triggered by changes in gut microbiota and neuroinflammation) are also thought to be involved in the onset of MDD [46,47,48]. The KYN pathway may be either neuroprotective through the production of kynurenic acid (KYNA, an NMDA receptor antagonist), or neurotoxic through the generation of quinolinic acid (QA, an NMDA receptor agonist), which can lead to glutamatergic excitotoxicity and ultimately neuronal damage [49,50,51]. Importantly, serotonin is synthesized through TRP catabolism [52], and evidence shows that inflammation triggers a shift from the TRP catabolism pathway to the neurotoxic kynurenine pathway, resulting in decreased serotonin production [53,54,55]. The enzyme kynurenine monooxygenase (KMO) is responsible for converting KYN into 3-hydroxy-kynurenine (3-HK), which is then further converted into QA. This enzyme is mostly expressed in macrophages, monocytes, and microglial cells [56,57]. On the other hand, the enzyme kynurenine aminotransferase (KAT) is mainly expressed by astrocytes, and is responsible for converting KYN to KYNA [49]. Therefore, the loss of astrocytes observed in MDD may compromise the synthesis of KYNA [58,59]. Moreover, QA-mediated excitotoxicity further result is astrocytic death in the MDD brain [51,60]. Indeed, an imbalance in the KYNA/QA ratio, with a reduction of KYNA and an excessive production of QA, has been observed in MDD patients [61,62,63].

## 3. Neuroinflammation: Function of Glial Cells

Neuroinflammation can be caused by numerous factors including stress, infections, autoimmune diseases, and gut dysbiosis. These factors induce morphological and functional alterations in glial cells, including microglia and astrocytes, and these are thought to play a fundamental role in the pathophysiology of MDD [64,65,66]. The main events underlying microglial and astrocytic activation during the neuroinflammatory process are shown in Figure 1.

### 3.1. Microglia

Microglia are the main immune cells present in the Central Nervous System (CNS). These cells are considered plastic, since they have different morphological and functional states that vary according to the conditions of the environment (homeostatic or pathological) [67]. Transcriptome analyses have revealed that during physiological conditions, microglia have a “resting” transcriptomic profile, which is characterized by greater expression of genes that contribute to CNS maintenance. In contrast, in models of neurodegeneration, inflammation, and ageing, microglia are thought to favor the expression of inflammatory markers, thus triggering a gradient of microglial activation [68,69,70]. For example, in a study using mice treated with lipopolysaccharide (LPS) in early life, it was possible to observe a series of alterations in microglial gene expression that favored the development of depressive-like behaviors in adolescence [71]. Furthermore, a sustained activation of microglia associated with an increase in the levels of pro-inflammatory mediators has been repeatedly observed in different models of depression induced by stress, inflammation or gut dysbiosis [72,73,74,75,76]. Moreover, alterations in microglia have been detected in the prefrontal cortex, anterior cingulate cortex, hippocampus and amygdala of MDD patients [77]. In addition, there appears to be a correlation between microglial activation and the severity of the depressive episode in humans [78,79].

Microglial cells express pattern recognition receptors (PRRs), such as Toll-like receptors (TLRs). TLRs mediate microglial activation in response to damage-associated molecular patterns (DAMPs) including ATP, heat shock proteins, high mobility group box 1 (HMGB1), RNA, and DNA, as well as pathogen-associated molecular patterns (PAMPs) such as bacterial lipoproteins, peptidoglycans, and endotoxins such as LPS [80,81]. Activation of these receptors through DAMPs and/or PAMPs induces morphological and genetic changes in microglia [82]. Morphologically, these changes are related to an increase in the size of the soma, retraction of processes, and reduced branching of the distal branches, resulting in an ameboid morphology [77,83].

During activation of TLRs, a pro-inflammatory signaling cascade is initiated [84]. For example, once activated by LPS, the TLR4 receptor associates with the adaptor protein myeloid differentiation factor 88 (MyD88) and induces the autophosphorylation of interleukin-1 receptor-associated kinase (IRAK). Phosphorylated IRAK1 and IRAK4 subsequently dissociate from MyD88, allowing it to interact with tumor necrosis factor receptor-associated factor 6 (TRAF6). This factor activates the transforming growth factor-β-activated kinase-1 (TAK1) complex, which promotes two inflammatory pathways involving mitogen-activated protein kinases (MAPK) and nuclear factor kappa-B (NF-kB) signaling [85,86,87]. In the NF-kB pathway, activated TAK1 complex induces the phosphorylation of the protein inhibitor of nuclear factor kappa B (IkB), which is then polyubiquitinated and degraded. This event allows NF-kB p50/p65 to move to the nucleus [88]. On the other hand, the activation of MAPK via TAK1 results in the phosphorylation and activation of the transcription factor activator protein-1 (AP-1) [85,87]. Both transcription factors bind to promoter regions that express pro-inflammatory genes such as TNF-α, inducible nitric oxide synthase (iNOS), cyclooxygenase-2 (COX-2), prostaglandin E2 (PGE2), IL-6, pro-IL-1β and components of the NLR family pyrin domain containing 3 (NLRP3) inflammasome [84,87,89]. In particular, the NLRP3 inflammasome as well as iNOS have been explored as potential therapeutic targets for the management of MDD [64,90], since inhibition and/or reduced expression of these proteins is related to improvements in depressive-like behaviors in animal models [91,92].

iNOS is an enzyme that catalyzes the synthesis of nitric oxide (NO) by converting L-arginine into L-citrulline and NO. Once synthesized, NO can be converted into nitrous anhydride and/or the reactive species peroxynitrite. In high concentrations, peroxynitrite may induce protein and lipid modifications, and the nitrosative deamination of DNA bases such as guanine and cytosine, thus impairing gene expression [93,94]. In addition to iNOS, the activation of neuronal nitric oxide synthase (nNOS) also may lead to deleterious effects eithin the CNS. Of note, an increase in the expression of nNOS has been shown to impair neuroplasticity [95,96]. Given this, overexpression or dysregulation of NOS is thought to contribute to MDD pathophysiology [90].

The continuous stimulation of microglia has been shown to promote the activation of the NLRP3 inflammasome, leading to autoproteolytic cleavage of pro-caspase-1 into active caspase-1. This can in turn cleave pro-IL-18 and pro-IL-1β into IL-18 and IL-1β, respectively [84,97]. In addition, NLRP3 activation can lead to gasdermin D-mediated membrane pore formation (GSDMD) and pyroptosis, characterized by intense cytokine efflux, swelling, and membrane disruption, which culminates in glial death and the release of DAMPs into the extracellular medium [84]. DAMPs released in this process can further contribute to the maintenance of microglial activation. For example, ATP and/or ADP can activate P2X7 purinergic receptors, which release TNF-α, thus contributing to the neuroinflammatory process in MDD [98,99].

It is important to note that under neuroinflammatory conditions, microglia tend to express fewer neuroprotective genes, which under normal circumstances provide trophic support to neurons. These genes include BDNF, nerve growth factor (NGF), glial cell-derived neurotrophic factor (GDNF), and neurotrophins (NT) 4/5. A decrease in trophic support can in turn impair synaptic plasticity-related mechanisms, including neurogenesis [100]. Further to this, it has also been shown that activation of microglia CX3CR1 receptors via CX3CL1 (a CX3C class chemokine originating from neurons) also contributes to neuronal survival [101]. However, under neurotoxicity conditions, there is a decrease in CX3CL1 levels, further favoring an inflammatory state [102,103]. For example, a study with mice subjected to a model of LPS-induced depressive behavior reported a decrease in CX3CL1 levels and microglial activation and an associated increase in pro-inflammatory cytokines [104]. Considering that mice deficient in CX3CR1 have impaired neurogenesis and synaptic plasticity, reduced expression of this chemokine could negatively affect neuroplasticity pathways [105]. Interestingly, the neuronal glycoprotein cluster of differentiation 200 (CD200) acts in a similar way to CX3CL1 by activating the microglial CD200R1 receptor [106,107]. In a chronic stress model induced by social defeat, the induction of depressive-like behaviors was related to microglial activation and a decrease in CD200 in the hippocampal dentate gyrus. In contrast, exogenous CD200 treatment alleviated the neuroinflammatory response and increased BDNF expression, which in turn improved hippocampal neurogenesis in this model [107]. These findings suggest that microglial activation is one of the possible mechanisms underlying the impaired neuroplasticity seen in MDD [77].

### 3.2. Astrocytes

In addition to microglia, astrocytes are also glial cells that can undergo morphological, molecular, and functional changes in response to inflammation, turning into “reactive” astrocytes. These changes alter the ability of astrocytes to maintain CNS homeostasis, compromising neuronal survival [108]. In addition, the phagocytic activity and the ability to modulate excitatory synapses also appear to be compromised in reactive astrocytes [109]. Over the last few years, several studies have linked astrocytic activation and MDD [110,111,112]. In line with this hypothesis, an increase in plasma levels of markers of astrocytic activation, such as glial fibrillary acidic protein (GFAP) and S100β, are here shown to be significantly increased in patients with treatment-resistant depression when compared to healthy individuals [113].

Transcriptome studies have revealed that during adverse conditions, such as aging and inflammation, there is a change in astrocytic gene expression. Under these circumstances, the expression of homeostatic markers is decreased, while the expression of genes related to pro-inflammatory mechanisms is enhanced. Interestingly, these changes appear to be dependent on the brain region where these cells are located [70,114]. Interestingly, transcriptome analysis indicated that the induction of inflammatory gene expression in astrocytes is dependent on the Orai1 calcium channel, since genetic inhibition of this channel prevented this induction. In addition, the knockout of Orai1 prevented an increase in the levels of inflammatory mediators in the hippocampus and the occurrence of depressive-like behavior in LPS-treated mice [112].

Of note, rodent astrocytes do not express TLR4, which is necessary for LPS recognition. In humans, although astrocytes express this receptor, they lack its downstream pathways, including MYD88 [108]. As such, astrocytic activation seems to be partly dependent on microglial activation. Indeed, activated microglia have been shown to secrete mediators such as IL-1α, TNF-α, and complement component 1q (C1q), which can induce the activation of astrocytes and consequently promote the synthesis and release of a neurotoxin that favors caspase 2/3-mediated apoptosis in oligodendrocytes and mature neurons. A recent study reinforced these findings by demonstrating that activation of astrocytes in a stress-induced depression model is only dependent on the activation of the NLRP3 inflammasome present in microglia since the specific knockout of astrocytic NLRP3 was unable to mitigate the activation of these glial cells [115]. Another mechanism by which microglia can induce astrocytic activation is through stromal cell-derived factor (SDF)-1a, which interacts with and activates CXC chemokine receptor type 4 (CXCR4). The activation of these receptors results in a series of events, which favor a substantial increase in the release of glutamate by astrocytes, inducing glutamatergic excitotoxicity [116,117].

It is well known that neuroinflammation induces glutamatergic dysfunction in astrocytes, and glutamatergic excitotoxicity has been associated with MDD [118,119]. Several mechanisms can result in an increase in excitotoxicity, including dysfunction of glutamate transporters, malfunction of glutamatergic receptors (particularly NMDA receptors), and alterations in glutamate and calcium metabolism. In addition, factors such as mitochondrial dysfunction, neuronal damage, and oxidative stress can all favor excitotoxicity. These mechanisms promote a substantial increase in glutamate in the synaptic cleft, leading to intense activation of glutamatergic receptors [120]. Initially, ionotropic receptors such as AMPA receptors, kainate receptors (KAR), and NMDA receptors are hyperactivated, culminating in an intense influx of sodium (and calcium). Subsequently, metabotropic glutamate receptors (mGluRs) are activated and synthesize second messengers such as diacylglycerol and inositol 1,4,5-triphosphate, which can then activate downstream signaling pathways [120,121]. The activation of glutamate ionotropic receptors favors neuronal swelling, mainly due to the influx of sodium, and disrupts the ionic gradients present across mitochondrial membranes and the endoplasmic reticulum, culminating in the release of calcium from these intracellular organelles [122]. The resulting significant increase in intracellular calcium can then activate enzymes that degrade proteins, lipids, and nucleic acids. Other enzymes can also be activated, such as phospholipase A2, cyclooxygenase-2, and lipoxygenases involved in the synthesis of arachidonic acid, as well as its conversion into prostaglandins, leukotrienes, and thromboxanes, a process that leads to the concomitant production of reactive oxygen species (ROS). In addition, it has been shown that calcium-induced activation of phospholipase A2 is capable of preventing neuronal hyperpolarization by inhibiting gamma-aminobutyric acid (GABA) receptors (GABARs), further contributing to glutamatergic excitotoxicity [120,123,124].

Notably, an increase in intracellular calcium concentration also promotes hyperactivation of sodium/potassium and calcium ATPases as a measure to neutralize ion influx. However, this results in excessive ATP consumption, resulting in neuronal energy depletion. This state is further exacerbated by mitochondrial dysfunction, which is also caused by excess calcium. Indeed, excessive mitochondrial calcium uptake through a specific mitochondrial calcium transporter leads to mitochondrial depolarization, impairing ATP synthesis as well as mitochondrial antioxidant defenses, further exacerbating energy depletion and resulting in ROS formation. Together, these mechanisms eventually culminate in apoptotic neuronal death through the activation of caspases and calpains [120]. In addition, excess extracellular glutamate can also inhibit the xc- system (a cystine/glutamate antiporter that captures cystine inside the neurons and releases glutamate into the extracellular environment), resulting in ferroptosis, another form of cell death [125,126].

Furthermore, some enzymes associated with the cytosolic tail of the NMDA receptor, such as calpains, death-associated protein kinase 1 (DAPK1), and nNOS are directly activated after the stimulation of this glutamate receptor [122]. In contrast, cAMP Response Element-Binding Protein (CREB) signaling, which is known to promote neuroplasticity mechanisms, is negatively regulated by this receptor [127].

### 3.3. Changes in the Blood–Brain Barrier Due to Glial Activation

Neuroinflammation has been associated with alterations in the permeability of the blood–brain barrier (BBB) [128]. Indeed, microglial and astrocytic activation results in the synthesis and release of numerous chemokines and cytokines, including monocyte chemoattractant protein-1 (MCP-1), macrophage inflammatory protein1α (MIP-1α), IL-6, TNF-α, and IL-1β, which impair the integrity of the BBB [129]. In addition to pro-inflammatory mediators, activated glial cells significantly lead to the production of ROS and reactive nitrogen species (RNS), which can in turn contribute to an increase in barrier permeability [130]. Particularly, ROS synthesized by microglia can activate signaling pathways that induce the activation of matrix metallopeptidase (MMP)-9, MMP-3, and MMP-2, which remodel the cytoskeleton and negatively regulate the expression of tight junctions, claudins, occludin, and zona occludin (ZO) [129].

In particular, IL-1β synthesized by microglia and activated astrocytes is known to contribute to increased BBB permeability by interacting directly with IL-1 receptor type 1, which is expressed by BBB cells, including brain microvascular endothelial cells, perivascular astrocytes, and microglia [131]. In addition, IL-1β activation induces an increase in the transcription of various adhesion molecules in brain microvascular endothelial cells, and these help activated leukocytes adhering to the surface of these cells [129,132,133]. Particularly in astrocytes, the expression of vascular cell adhesion molecule-1 (VCAM-1) dependent on TNF receptor 1 (TNFR1) has been associated with the traffic of T cells from the perivascular spaces to the parenchyma [134]. In addition to IL-1β, other mediators such as IL-6, CXCL10, and chemokine ligand 2 (CC motif) (CCL-2), as well as chemokine receptors 7 and 8 (CCR7 and CCR8), are also able to recruit dendritic and peripheral immune cells, contributing to the impairment of BBB integrity [135,136,137,138].

Of particular interest, it has been shown that patients with MDD have an impaired BBB [139]. For example, a decrease in the expression of claudin-5 (a protein that maintains the integrity of a paracellular aqueous channel present in brain endothelial cells, thus regulating flow of ions and peripheral cytokines) has been reported in the hippocampus of individuals afflicted with MDD [140]. Moreover, several markers of endothelial dysfunction, including intercellular adhesion molecule-1, VCAM-1, E-selectin, and von Willebrand factor, have also been detected in individuals that developed MDD later in life [141].

## 4. Role of the Peripheral Immune System in MDD

Several lines of evidence have also proposed a relationship between the peripheral immune system and the neuroinflammatory process [142,143,144]. The immune system is composed of innate and adaptive systems. The innate immune system consists of myeloid cells including macrophages, monocytes, dendritic cells, and lymphoid cells such as natural killer (NK) cells, which act as first defense by rapidly responding to pathogens, for example. The adaptive immune system is formed by T and B lymphocytes, which act more slowly, as they require prior activation and differentiation steps to be able to carry out their functions. Specifically, B cells proliferate and differentiate into plasma cells, which then produce specific antibodies. On the other hand, activated T cells can differentiate into cytotoxic, helper, and regulatory T cells. Cytotoxic T cells (CD8^+^ cells) kill infected cells, whereas helper T cells (Th) modulate the activity of other immune cells, and regulatory T cells (Tregs) suppress the activity of other lymphocytes to prevent autoimmunity [145]. These cells can contribute to neuroinflammation by entering the CNS via fenestrations in the BBB and/or through the circumventricular sensory organs located in the walls of the third and fourth ventricles. These regions are composed of fenestrated capillary loops surrounded by large perivascular spaces, which facilitate the flow of cytokines and immune cells from the periphery to the brain [146].

In certain MDD patients, it is possible to observe an altered peripheral immune system profile, with an increase in the levels of pro-inflammatory cytokines, including IL-1, IL-6, TNF-α, and IL-1β [147]. Peripheral or local administration of cytokines and endotoxins in animals, as well as animal models based on exposure to stress, also reinforce the finding that prolonged peripheral immune activation can trigger a neuroinflammatory process and result in depressive-like behaviors [148,149,150]. In particular, chronic exposure to stress was shown to result in the dysfunction of the HPA axis, which in turn can induce various pathophysiological changes. Furthermore, an impairment in the function of glucocorticoid receptors has also been associated with MDD [151]. Among their several functions, these receptors have an immunosuppressive effect by inhibiting the translocation of NF-kB to the nucleus [152,153]. Moreover, chronic stress favors an increase in norepinephrine (noradrenaline), which acts on adrenergic receptors present in immune cells, thus positively regulating the transcription of several pro-inflammatory genes, including IL-1, IL-6, and TNF-α [154,155]. Changes in immune cell functions are also observed after exposure to chronic stress, including impaired migration of cells to the inflammatory site, decreased cytotoxic activity of NK cells, and decreased levels of total T lymphocytes and circulating Th lymphocytes [154]. Furthermore, MDD has also been associated with reductions in the levels and function of both T and NK cells [156].

Of note, IL-6 can enhance Th17 cell differentiation, thereby causing an imbalance between Th17 and Treg cells [34]. In line with this, a greater propensity for Th17 differentiation has been observed in individuals with MDD [157]. Likewise, in animal models of depression, an increase in brain levels of Th17 has also been reported [158,159]. Th17 cells are CD4^+^ Th lymphocytes that secrete large amounts of IL-17A [160], with IL-17A being able to induce cytokine secretion and glial cell stimulation, thus enhancing neuroinflammatory responses [160,161]. Furthermore, decreased levels of Treg and its regulatory cytokine IL-2 have also been associated with MDD [162,163,164]. Indeed, treatment with low doses of IL-2 is able to attenuate depressive-like behavior by restoring Treg levels and reducing the neuroinflammation state in a stress model [34]. Indeed, several classic antidepressants, as well as ketamine, have been shown to restore Treg levels and decrease Th17 levels [165,166,167], which may contribute to their antidepressive properties. As such, compounds that decrease Th17 and/or increase Treg levels may be of potential therapeutic value for the treatment of MDD.

The role of the peripheral immune system in MDD is further reinforced by studies showing an association between several autoimmune diseases with this psychiatric disorder [168,169]. Indeed, individuals with MDD appear to have higher levels of reactive antibodies, which suggests that this disorder may be a risk factor for various immune system-related diseases [170]. In agreement with this hypothesis, a prospective population-based study found that patients with MDD were at higher risk of developing rheumatoid arthritis, psoriasis vulgaris, systemic lupus erythematosus, multiple sclerosis, Crohn’s disease, and type 1 diabetes [171]. It is noteworthy that higher levels of inflammatory markers are also related with decreased responsiveness to antidepressant treatment [172].

While several studies have highlighted the importance of inflammatory mechanisms in adults with depression [7,173], discrepant results have been reported in youth. During adolescence and young adulthood, the immune system undergoes various alterations, including a reduction in lymphatic tissue size and changes in sex hormones, which can in turn affect cytokine release [174]. These physiological changes seen during this period may account for the inconsistent findings that have been reported in the literature, and further research is warranted in order to ascertain how inflammation contributes to MDD in youths.

A few systematic reviews have attempted to examine the relationship between inflammation and depression in youths [12,175,176]. In a systematic review conducted by D’Acunto et al. [176], only a trend for significantly higher levels of peripheral TNF-α was observed in youths with depression, when compared to healthy controls. Toenders et al. [175] identified 109 studies examining the association between inflammation and depression in youths, and determined that adolescents with depression showed higher level of IL-1β when compared to healthy controls [175]. However, no significant differences in the levels of other cytokines were observed [175]. In longitudinal studies, it is possible to show that higher baseline IL-6 levels and changes in TNF-α levels were predictive of an increase in depressive symptoms in youths at follow-up [175]. Similarly, the most recent systematic review conducted in 2020 by Colasanto et al. [12] included twenty-two studies (20,791 participants) and showed a significant association between concurrent depression and CRP and IL-6 levels. In addition, longitudinal analyses revealed that depression is a significant predictor of IL-6 and, conversely, inflammatory markers (CRP or IL-6) predict future depression [12].

It is also important to note that differences in inflammation between sexes may also be related with the fact that the incidence of depression is higher in females than males. Differences in sex hormones between males and females, and changes in hormone levels during puberty, menstruation, pregnancy, and menopause may account, at least in part, for the different incidence of this mood disorder between females and males [177]. Male sex hormones have mainly anti-inflammatory activity, while female sex hormones have both pro- and anti-inflammatory activities. Moreover, females have more innate and adaptive immune cells, higher inflammatory marker levels, and higher risk of developing autoimmune disorders in comparison with males [177,178]. In agreement, several studies have demonstrated that females with MDD appear to have an increase in serum levels of IL-1β, IL-6, IL-8, TNF-α, IFN-γ, and CRP [179,180,181,182,183]. Conversely, males with MDD appear to have an increase in serum levels of TNF receptor 2 and IL-17 [183].

Overall, the evidence suggests that an activated inflammatory response system (IRS) contributes to the pathophysiology of MDD. This IRS is characterized by microglial, monocytic, and lymphocytic activation, which culminates in the synthesis of inflammatory mediators, including TNF-α, IL-1β, IL-6, soluble IL-6 receptor (sIL-6R), IFN-γ, IL-2, and IL-17 [184,185]. However, some studies have also shown that some patients with depression exhibit increased Th2 and Treg activity, suggesting the presence of a compensatory immune response system (CIRS), which is characterized by increased levels of anti-inflammatory cytokines such as IL-4 and IL-10, as well as increased levels of soluble cytokine receptors (sIL-2R, sTNF-R1, sTNF-R2) and of the soluble IL-1 receptor antagonist (sIL-1RA) [185,186,187,188]. Indeed, some patients with depression have increased levels of both pro- and anti-inflammatory cytokines [188,189]. Therefore, components of CIRS probably counteract the effects of IRS in the context of MDD. The CIRS/IRS imbalance, in turn, could be a key factor in the development of a chronic inflammatory response observed in some MDD patients, particularly those who are resistant to treatment with conventional antidepressants [184]. In view of this, several studies have assessed the levels of these mediators, so as to better understand the relationship between CIRS/IRS in MDD. For example, studies included in a meta-analysis found an association between increased levels of IL-1RA, IL-6, IL-10, IL-12, sIL-2R, sIL-6R, and TNF-α, and decreased levels of IFN-γ and IL-4, in adult patients with depression [190]. However, the levels of these markers seem to vary between studies, especially considering variables such as age and sex. In a recent study conducted with adolescents with depression, it was possible to see a significant increase in some markers such as IL-4 and Treg + Th2, which have not been observed in most studies conducted with adults. Furthermore, when the analyses considered potential sex differences, it was found that female adolescents with MDD only had increased levels of IL-10 and TNF-α, while male adolescents with MDD had increased levels of IL-4, IL-10, sIL-6R, Treg + Th2, and TNF-α/TNF-R1 [191]. Finally, a study conducted by Sowa-Kućma et al. [192] reported that the severity of MDD, measured with the Hamilton Depression Rating Scale, was correlated with an increase in IRS and CIRS markers, including sIL-6R, tumor necrosis factor receptor 80kDa (sTNFR80), and zCytR (z-unit weighted indices reflecting the 5 cytokine receptor levels). This study also showed that previous suicide attempts are associated with increased sIL-1RA and IL-1α levels [192].

## 5. Gut Microbiota

Gut microbiota has been recognized as being crucial for mental health. This comprises a variety of microorganisms that include bacteria, archaea, protozoa, fungi, and algae, which play an important role in gut physiology and homeostasis [193]. Recently, the bidirectional connections between the gut microbiota and the brain, known as the microbiota-gut-brain axis, has gained greater interest, especially in the context of psychiatric illnesses, including MDD [194]. This connection between gut and brain occurs through the autonomic nervous, enteric nervous, neuroendocrine, and immune systems, and involves microbial-derived metabolites, chemical molecules, and neuronal pathways [185].

In a state of eubiosis, in the presence of a healthy gut microbiota, these microorganisms produce several secondary metabolites. Bacteria can ferment indigestible dietary fibers to produce short-chain fatty acids (SCFAs), mainly acetate, propionate, and butyrate, which help to maintain the integrity of the gut barrier, mucous production, and control inflammation [195,196]. Notably, SCFAs can enter the bloodstream, cross the BBB, and reach the brain, where they can exerce anti-inflammatory and pro-neurogenic effects, modulate neurotransmitter systems including the glutamatergic, GABAergic, serotonergic, dopaminergic, adrenergic, and noradrenergic systems, and maintain BBB integrity [195,196,197,198]. Gut microbiota is also capable of producing various neurotransmitters (GABA, serotonin, dopamine, and norepinephrine) and polyamines, which can then act both in the periphery and within the CNS [199,200].

Gut microbiota are thought to play important roles in the regulation of innate and adaptative immune responses at the level of mucosal surfaces, protecting them against inflammation, infection, and autoimmunity [201]. Gut microbiota also contribute to priming and activating immune mediators, regulating their development and function in various organs, including the brain [201,202]. In addition, together the gut microbiota, gut epithelium, and gut immune system can regulate systemic inflammation, preventing commensal bacteria and pro-inflammatory molecules from crossing the gut barrier and reaching the bloodstream and the brain [201,203,204].

The vagus nerve, which transmits parasympathetic information from the brain to the gut and vice versa, constitutes the quickest and most direct link between these two organs. As such, this peripheral nerve provides the fastest way for gut microbiota and the immune system to directly influence the CNS [205]. In line with this hypothesis, a recent study conducted by Siopi et al. [206] has shown that the vagus nerve can indeed mediate the effects of gut microbiota on brain function and behavior. In this study, healthy mice received gut microbiota from mice submitted to chronic unpredictable mild stress (CUMS) and were shown to have vagus nerve-mediated changes in serotonin and dopamine pathways, which were associated with concomitant and persistent deficits in hippocampal neurogenesis and neuroinflammation. On the other hand, subdiaphragmatic vagotomy abolished the occurrence of depressive-like behaviors, neuroinflammation, and the deficits in hippocampal neurogenesis in these mice [206], providing further evidence to support the role of the vagus nerve as a primary communication pathway for gut microbiota-mediated neuroinflammation [207]. It is noteworthy that stimulation of the vagus nerve has been reported to attenuate neuroinflammation and reduce the release of pro-inflammatory factors, suggesting that vagal stimulation may have antidepressant effects [208].

Imbalances in gut microbiota homeostasis, including alterations in microbiota composition and specific taxa, can lead to gut dysbiosis, resulting not only in peripheral inflammation, but also in neuroinflammation, reduced BBB integrity, neuronal death, microglia dysfunction, and depressive symptoms [194,204,209]. Accordingly, recent studies have shown differences in the gut microbiota of MDD-afflicted individuals as compared to healthy controls [210,211,212,213,214,215,216,217,218].

A disruption in the gut barrier and a consequent increase in its permeability, commonly referred to as a “leaky gut”, may be caused by several stressors, including environmental factors, immune factors such as pro-inflammatory cytokines, and gut-microbiota-related factors including microbiota dysbiosis [81]. Chronic intestinal inflammation leads to a “leaky gut”, allowing pro-inflammatory cytokines, bacterial endotoxins (such as LPS), metabolic bacterial components, and immunoglobulins to overflow from the intestine and into the bloodstream, eventually reaching and compromising BBB integrity, and entering the brain, where they can promote neuroinflammation [219]. In line with this, BBB integrity appears to be affected in individuals with MDD, as illustrated by a clinical study showing increased serum levels of S100B (a marker associated with BBB damage) in individuals with MDD when compared with healthy controls [220]. A relationship between gut microbiota and BBB integrity and permeability has also been illustrated through preclinical studies. For example, mice that were germ-free since fetal development were shown to display increased BBB permeability when compared to pathogen-free mice with an healthy gut microbiota [221]. Rhesus monkeys that received antibiotic treatment presented an altered microbiome and showed an increase in BBB permeability [222]. Furthermore, exposure to a low dose of penicillin during the late prenatal period and early postnatal life resulted in changes in gut microbiota, an increase in cytokine expression in the pre-frontal cortex, altered BBB integrity, and behavioral changes [223].

Indeed, there is a noticeable connection between persistent gut inflammation and MDD. The prevalence of MDD symptoms in individuals with inflammatory bowel disease (IBD) is estimated to be approximately 21% to 25.2%, and the prevalence of clinically diagnosed MDD within this population is approximately 15.2% [224]. In addition, a “leaky gut” and increased LPS translocation have also been reported in individuals with MDD [225]. Data from animal models of colitis have also supported the close relationship between gut inflammation and mood [226,227,228]. For example, Yoo et al. [227] showed that fecal microbiota transplantation (FMT) from patients with IBD (Crohn’s disease and ulcerative colits) plus MDD caused depressive-like behaviors in mice, while also increasing peripheral, colonic, and hippocampal levels of corticosterone, IL-1 β, IL6, LPS, and decreasing IL-10 levels. Conversely, FMT from healthy donors or sulfasalazine treatment alleviated the depressive-like behaviors induced by FMT from IBD plus MDD patients, while also reducing pro-inflammatory markers in the serum and colon of these mice [227]. In addition, administration of *Lactobacillus plantarum* NK151, *Bifidobacterium longum* NK173, and *Bifidobacterium bifidum* NK175 alleviated depression-like behaviors induced by FMT from IBD plus depression patients, while also normalizing hippocampal NF-κB^+^Iba1^+^ cell numbers, IL-1β and IL-6 expression, serum levels of LPS, IL-6, and creatinine, as well as colonic NF-κB^+^CD11c^+^ cell numbers and IL-1β and IL-6 expression in these mice [218]. Other preclinical studies have also evaluated colitis and the occurrence of depressive-like behaviors in dextran sulfate sodium (DSS)-treated mice [226,228]. Takashi et al. [226] show that DSS-treated mice presented higher TNF-α and IL-6 expression in the rectum and hippocampus, activated caspase-3 in the hippocampus, as well as decreasing hippocampal neurogenesis, and these changes were reversed by administration of *Enterococcus faecalis* 2001, which also prevented the occurrence of DSS-induced depressive-like behaviors. In a more recent study, Wadie et al. [228] showed that niacin ameliorated DSS–induced behavioral deficits, alleviated macroscopic and microscopic colonic inflammatory changes, restored BBB integrity through enhancement of ZO-1, occludin, and claudin-5 protein levels in the hippocampus, while also decreasing IL-1β and NF-kB levels and increasing glutathione (GSH), sirtuin-1 (Sirt-1), nuclear factor erythroid 2–related factor 2 (Nrf2), and heme oxygenase-1 (HO-1) concentrations in the hippocampus. Furthermore, 2,4,6-trinitrobenzenesulfonic acid (TNBS)-induced colitis was shown to activate NF-kB, increase gut permeability, fecal and blood levels of LPS, and the number of Enterobacteriaceae (particularly *Escherichia coli*) in the gut microbiota in mice. Of note, these changes were accompanied by memory impairments. On the other hand, administration of *E. coli* isolated from the feces of mice with TNBS-induced colitis caused memory impairment and colitis in mice [229]. Importantly, in both cases (colitis induced with either TNBS or *E. coli*) an increase in NF-kB activation and TNF-α expression, as well as suppression of *bdnf* expression, were observed in the in the hippocampus of mice [229].

In conclusion, gut microbiota appears to play a fundamental role in homeostasis and the control of both peripheral and central inflammation, and may constitute a relevant therapeutic target for psychiatric diseases, including MDD.

## 6. Anti-Inflammatory Properties of Antidepressants

The anti-inflammatory effects of antidepressant drugs have been robustly demonstrated in numerous studies. For example, tricyclic antidepressants, including citalopram, clomipramine, and imipramine were shown to inhibit the release of IL-6, IL-1β, and TNF-α in human monocytes [230]. The anti-inflammatory effect of clomipramine has been corroborated in LPS-treated C57BL/6 mice. In this model, clomipramine (20 mg/kg for 1 week, i.p.) was able to decrease the expression of IL-1β, IL-6, and TNF-α. However, although cytokine synthesis, particularly IL-1β, is known to be related with the NLRP3 inflammasome, no changes were observed with regard to the expression of proteins that make up this inflammasome in this model [231]. In agreement, a different study in which the same dose of clomipramine (20 mg/kg for 1 week, i.p.) was administered to C57BL/6J male mice treated with LPS also failed to detect any changes in the immunocontent of NLRP3 and caspase-1 [232]. In contrast, in BV2 cells, clomipramine was effective in attenuating NLRP3 expression induced by LPS [231]. Other tricyclic antidepressants, such as amitriptyline and doxepin (both administered at a dose of 10 ng/mL for 2 h) were able to inhibit microglial activation in vitro [233]. However, in a different study, amitriptyline (20 or 50 μM for 1 h) was unable to mitigate caspase-1 activation after stimulation of mouse glial cells with LPS/ATP or LPS/nigericin [234]. Given these discrepancies found in the literature, the exact anti-inflammatory mechanisms activated by tricyclic antidepressants warrant further exploration.

Anti-inflammatory mechanisms also underlie the effects of fluoxetine, the most widely prescribed antidepressant in the clinical setting. Indeed, it has been shown that treatment with fluoxetine significantly reduces the levels of IFN-γ, TNF-α, and the IFN-γ/IL-10 ratio in the blood of normal volunteers [235]. Several pre-clinical studies have also reinforced the anti-inflammatory properties of this antidepressant. In male Wistar rats exposed to a stress model, treatment with fluoxetine (10 mg/kg for 6 weeks, i.p.) was able to inhibit the activation of the NF-kB pathway, compromising the expression of proteins that constitute the NLRP3 inflammasome, and, consequently, the levels of IL-1β in the prefrontal cortex [236]. One of the mechanisms that may be related to the inhibition of the NF-kB pathway by fluoxetine is the increase in the levels of the NF-kB inhibitor, IkB-α. Indeed, this drug was shown to interact directly with IkB-α, preventing its ubiquitination in an in vitro model of cerebral ischemia/reperfusion [237]. Furthermore, fluoxetine treatment (10 mg/kg for 4 weeks) also suppressed the activation of the NLRP3 inflammasome and the subsequent cleavage of caspase-1 and secretion of IL-1β in hippocampal microglia from male C57BL/6 mice subjected to stress [238]. In a model of LPS-induced depression, fluoxetine (20 mg/kg, i.g.) inhibited glial activation, decreasing levels of pro-inflammatory cytokines such as TNF-α, IL-1β, and IL-6, and increasing levels of the anti-inflammatory cytokine IL-10 in the hippocampus of male C57BL/6J mice [239]. In agreement with these findings, the role of fluoxetine in inhibiting astroglial and microglial activation has been further reinforced by other studies [75,240]. Fang et al. [240] showed that fluoxetine can inhibit astrocytic activation via the 5-HT_2B_R/β-arrestin2 pathway in both in vivo and in vitro models of depression. More recently, fluoxetine was also shown to positively regulate the expression of nuclear receptor-related protein 1 (Nurr1) and FosB (FosB Proto-Oncogene, AP-1 Transcription Factor Subunit), inhibiting inflammation-induced morphological and functional alterations in microglia in the anterior cingulate cortex [75].

Anti-inflammatory properties have also been associated with the antidepressant effect of ketamine. For example, a subanesthetic dose of ketamine (10 mg/kg, i.p.) was able to attenuate the expression of IL-1β and NLRP3 in the hippocampus of male C57BL/6 mice treated with LPS [241]. Similar effects were also observed with pre-treatment of this drug. Indeed, a single dose of ketamine (5 mg/kg, i.p.) administered 1 week before treatment with TNF-α and LPS had a protective effect in mice, preventing an increase in the components of the NLRP3 inflammasome complex [149]. Although a few discrepancies can be found in clinical studies, some have described supported the anti-inflammatory action of ketamine [242]. For example, ketamine infusion (0.5 mg/kg, i.v., single dose) was able to decrease serum TNF-α levels in patients with treatment-resistant MDD [243]. In another study, the same dose of ketamine was also able to reduce serum IL-6 levels in MDD-afflicted individuals [244]. Together, these data suggest that anti-inflammatory mechanisms may indeed contribute to the effects of antidepressant drugs.

## 7. Antidepressant Effects of Anti-Inflammatory Drugs

Given the strong relationship between MDD and inflammation, a series of studies has evaluated whether anti-inflammatory drugs could exert antidepressant effects. Notably, the inflammatory cascade triggered in the brain can culminate in the induction of COX. Therefore, it has been suggested that inhibiting this enzyme could be beneficial in depression [245]. In agreement, preclinical studies have shown beneficial effects of selective COX-2 inhibitors such as celecoxib in models of depression [246,247]. For example, treatment with celecoxib (16 mg/kg for 21 days) was effective in alleviating the depressive-like behavior of rats submitted to a chronic stress model [246]. A similar effect was also observed in another model of depression. In mice subjected to chronic inflammation induced by Complete Freund’s Adjuvant, co-treatment of celecoxib and bupropion (3 mg/kg) inhibited the occurrence of depressive-like behaviors. It is noteworthy that a higher dose of celecoxib per se (30 mg/kg) was also effective [248]. In a recent study, celecoxib (10 or 20 or 30 mg/kg) and cannabidiol (30 mg/kg) also showed a synergistic effect in models of depression induced by LPS or chronic social defeat stress. Interestingly, when each compound was administered per se, only partial effects were detected [249].

Of note, recent systematic reviews have also concluded that celecoxib may have a beneficial effect on MDD in humans [250,251]. A systematic review based on 44 studies found that a dose of 400 mg/day used for 6 weeks is effective as a complementary treatment for MDD [251]. Although these results are promising, clinical studies have not yet been able to associate the antidepressant effects of celecoxib with the attenuation of inflammatory mediators. Therefore, future studies are necessary, especially to assess possible long-term adverse effects [251].

Studies conducted with non-COX selective drugs, such as ibuprofen, have provided mixed results [252,253]. For example, treatment with ibuprofen alone (40 mg/kg) or in combination with escitalopram (10 mg/kg) was reported to reverse the depressive-like behavior induced by chronic stress in rats [252]. On the other hand, it has been shown that ibuprofen (1 mg/mL) can attenuate the effects of antidepressants including citalopram, fluoxetine, and imipramine in behavioral tests such as the tail suspension test and the forced swim test [254]. In addition, ibuprofen (40 mg/kg) was unable to prevent the development of depressive-like behaviors in mice treated with LPS [253]. Divergent results have also been reported for acetylsalicylic acid. Some studies have shown that co-administration of acetylsalicylic acid (45 mg/kg) and fluoxetine (5 mg/kg) reversed the escape deficit seen in a chronic escape deficit model of depression [255,256]. In contrast, this drug (3 mg/mL) has been shown to attenuate the effects of citalopram [254]. A recent systematic review of preclinical and clinical studies found evidence indicating the safety and efficacy of low-dose acetylsalicylic acid in the treatment of mood disorders [257]. However, more long-term clinical trials are still needed to assess the efficacy and safety of these drugs in recurrent depression.

## 8. Alternative MDD Therapeutic Approaches with Anti-Inflammatory Properties

### 8.1. Physical Exercise

Physical exercise has been extensively studied for its role in the treatment and prophylaxis of MDD due to its ability to modulate several pro-neurogenic and synaptogenic pathways as well as regulating peripheral and systemic inflammation [258]. Preclinical and clinical studies have shown that one of the mechanisms underlying the effects of physical exercise on neuroinflammation is related to the modulation of the NLRP3 inflammasome pathway [259].

In a study conducted by Tang et al. [260], treadmill exercise alleviated post-stroke depression-related hippocampal damage through inhibition of NF-kB/NLRP3 inflammasome in mice. Similarly, Li et al. [261] showed that treadmill exercise mitigated the increase in TLR4, NLRP3, and NF-kB levels caused by post-stroke depression. Treadmill exercise was also shown to regulate hippocampal inflammation by reducing the number of NLRP3-, TLR4-, IL-1β-, and IL-10-positive neurons in the hippocampal CA1 region of mice subjected to CMUS as a model of depression [262], improve depression-like behaviors and decreasing levels of IL-1β, IL-18, NLRP3, cleaved caspase-1 P10, and CD11b in the hippocampus of ovariectomized mice [263], preventing the hippocampal increase of Iba-1, TXNIP, and activation of the NLRP3 inflammasome pathway induced by Aβ1−40 [264], and increasing the expression of SIRT3, while reducing levels of ROS, NLRP3, IL-1β, and IL-18 in the hippocampus of mice subjected to CUMS [265]. Furthermore, in a model of high-fat-diet-induced-obesity, treadmill exercise was able to reduce the content of IL-1β and NLRP3, promote the Nrf2/HO-1 pathway, increase BDNF content in the rat hippocampus [266], while also reducing the protein content of NF-kB, and upregulating the PI3K/AKT pathway in the rat prefrontal cortex [267]. In addition, treadmill exercise was activated in the PI3K/Akt/mTOR and AMPK/Sirt1 signaling pathways, while inhibiting the NF-kB/NLRP3/IL-1β signaling pathway in the hippocampus of rats with streptozotocin-induced type 2 diabetes mellitus [268].

Some clinical studies have also shown the ability of physical exercise to reduce inflammation by modulating the NLRP3 inflammasome pathway. A systematic review conducted by Ding and Xu [269] observed that regular exercise could significantly decrease IL-1β and IL-18 (i.e., the end products of the inflammasome pathway) in older adults. The same study also showed that aerobic exercise is the most effective training modality, and that low-to-moderate intensity and mixed intensity exercise elicited better effects in reducing IL-1β and IL-18 when compared to high-intensity exercise [269]. A reduction in P2X7 receptor activation (involved in NLRP3 inflammasome activation), NLRP3, and NF-kB mRNA expression was also seen in lymphomonocytes of athletes who performed physical activity at least 5 times/week and for more than 10h/week [270]. Similarly, chronic exposure to moderate intensity physical exercise was shown to reduce the NLRP3 gene expression as well as serum levels of IL-1β and IL-18 cytokines [271]. On the other hand, chronic exposure to high intensity physical exercise was associated with an increase in NLRP3 gene expression, as well as serum levels of IL-1β and IL-18 [271]. Thus, while low to moderate physical exercise appears to have anti-inflammatory effects, high intensity physical exercise has been associated with the induction of inflammation through an increase in circulating pro-inflammatory cytokines [272,273].

### 8.2. Agmatine

Agmatine is a member of the polyamine family, produced through the decarboxylation of L-arginine by the enzyme arginine decarboxylase [274]. A large number of preclinical studies have reported an antidepressant-like effect of agmatine in various animal models, including models of chronic corticosterone administration [275,276], stress [277,278,279,280,281], and inflammation [282,283].

In addition, agmatine has also been shown to exert anti-inflammatory effects and to modulate the NLRP3 inflammasome pathway. For example, agmatine was able to attenuate NLRP3 protein expression as well as caspase-1 and IL-1β mRNA and protein levels in the prefrontal cortex of rats exposed to CUMS [281]. In another study conducted by Sahin et al. [284], agmatine was shown to down-regulate the expression of NLRP3 inflammasome components (NLRP3, NF-kB, PYCARD, caspase-1, IL-1β, and IL-18) while also reducing levels of pro-inflammatory cytokine in the hippocampus, prefrontal cortex, and serum of rats submitted to an acute restraint stress protocol. Furthermore, agmatine also attenuated the increase in NLRP3, ASC, caspase-1, and IL-1β expression and immunocontent in the hippocampus, while reducing IL-1β serum levels in mice submitted to a pentylenetetrazol (PTZ) model of epilepsy [285].

Agmatine also appears to be able to attenuate inflammation by modulating other inflammatory pathways. For example, agmatine was able to mitigate the effects of a high-fat diet by increasing BDNF protein levels while reducing TNF-α and IL-6 protein levels in the rat hippocampus [286]. Similarly, agmatine was able to reduce TNF-α, IL-1β, IL-6, and IL-10 protein levels in the prefrontal cortex and hippocampus of mice with streptozotocin-induced diabetes [287]. In a rotenone-induced experimental model of Parkinson’s disease (PD), agmatine also reduced the levels of HMGB1, the receptor for advanced glycation end products (RAGE), TLR4, and of the proinflammatory cytokines TNF-α and IL-1β in the substantia nigra pars compacta (SNpc) of rats [288]. These results were further corroborated in a separate study, showing that the neuroprotective properties of agmatine in this rotenone model of PD are not only associated with a decrease in TNF-α and IL-1β, but also with a reduction in the levels of malondialdehyde (a marker of lipid peroxidation) and glial fibrillary acidic protein (GFAP, a marker of astrocytic activation) [289]. The anti-inflammatory properties of agmatine have also been demonstrated through in vitro studies, where this compound was able to suppress protein expression of TLR4, MYD88, phospho-IkBα, phospho-NF-kB, and the NLRP3 inflammasome [285], while inducing an anti-inflammatory phenotype in BV-2 microglial cells treated with LPS [290]. Similarly, agmatine was shown to suppress nitrosative and oxidative stress and NF-kB expression, while stimulating the antioxidant Nrf2 pathway, thus resulting in decreased TNF, IL-1β, and IL-6 release, as well as reduced iNOS and COX-2 levels in LPS-stimulated BV-2 microglial cells [291]. Finally, agmatine was also shown to attenuate cell death and the expression of IL-6, TNF-α, and CCL2 in Neuro2A cells under high glucose conditions [292].

### 8.3. Probiotics

Considering the important role of the gut microbiota in maintaining homeostasis and the pathophysiology of MDD, probiotics have been proposed as a promising therapeutic approach for the treatment of MDD and other psychiatric disorders [293]. Probiotics are living microorganisms known for their beneficial modulatory effect on gut microbiota when ingested in adequate quantities [294]. Of note, modulation of the immune system has been proposed as one of the main mechanisms responsible for the beneficial effects of probiotics [295]. In agreement with this hypothesis, prolonged 21-day intake of the probiotic *Bifidobacterium adolescentis* was shown to prevent depressive-like behaviors and to decrease the expression of IL-1β, TNF-α, NF-kB, and Iba1 while increasing the expression of BDNF in the hippocampus of mice submitted to chronic restraint stress [296]. In a different study, pretreatment with *Clostridium butyricum* strain Miyairi 588 was shown to decrease IL-6 and IL-1β levels in the colon and hippocampus of mice submitted to chronic social defeat stress. In addition, this probiotic also reduced microglial activation, improved depressive-like behaviors, and enhanced gut levels of the Firmicutes phylum [297].

The antidepressant effects of probiotics have also been demonstrated in clinical trials. A randomized, double-blinded, placebo-controlled trial conducted in Iran with MDD patients showed that treatment with the probiotics *Lactobacillus acidophilus*, *Lactobacillus casein*, and *Bifidobacterium bifidum* improved depressive symptoms’ total scores, while also reducing inflammation by decreasing insulin resistance and serum CRP levels when compared to placebo [298]. Another randomized controlled trial found that eight weeks of treatment with *Lactobacillus helveticus* and *Bifidobacterium longum* resulted in a significant decrease in depressive symptoms, which was accompanied by a reduction in the serum KYN/TRP ratio [299], suggesting an effect of probiotics on TRP metabolism, which, as mentioned above, is closely related to inflammatory mechanisms and is implicated in the neurobiology of MDD.

### 8.4. Vitamin C (Ascorbic Acid)

Ascorbic acid, also known as vitamin C, is a water-soluble micronutrient required for multiple biological functions, acting mainly as a potent antioxidant [300]. It is also present in high concentrations in the brain (2–10 mM), where it exerts neuroprotective and neuromodulatory properties [301,302,303]. Ascorbic acid can pass from the bloodstream to the cerebrospinal fluid through the choroid plexus in the form of an ascorbate anion. In addition, it can also cross the BBB in the form of dehydroascorbate, being subsequently taken by neurons and glial cells [304].

Severe ascorbic acid deficiency has been linked to psychiatric disorders including MDD, and, in some patients, depressive symptoms precede the physical symptoms of scurvy [305,306]. Indeed, low levels of ascorbic acid have been associated with the increased incidence of depressive symptoms in humans [307], and several studies have shown that ascorbic acid can elicit antidepressant-like effects in animal models [308,309,310,311,312]. Although the exact mechanisms underlying its antidepressant effect are not yet well established, they appear to depend on its pleiotropic and anti-inflammatory activity. In agreement, ascorbic acid was able to prevent TNF-α-induced depressive-like behaviors in an inflammatory mouse model of depression, and this effect seemed to be associated with a reduction of p38MAPK phosphorylation, modulation of monoaminergic and glutamatergic systems, and nitric oxide synthesis [313]. Furthermore, ascorbic acid was also able to improve depressive-like behaviors, hyperglycemia, and hypoinsulinemia, while also increasing the levels of monoamines, decreasing oxidative stress, and reducing the levels of TNF-α and IL-6 in the prefrontal cortex of a rat model of type 2 diabetes and comorbid depression [314]. While further studies are still warranted so as to better understand the mechanisms involved in the antidepressant effects of ascorbic acid, the data discussed above suggest that, at least in part, these effects are dependent on its anti-inflammatory properties.

### 8.5. Vitamin D

In the context of MDD, the antidepressant-like properties of vitamin D have been shown in several pre-clinical animal studies [315,316,317,318]. However, only a limited number of studies has explored the anti-inflammatory mechanisms underlying its antidepressant action. For example, treatment with vitamin D3 (2.5 μg/kg, p.o., for 7 days) has been shown to induce an antidepressant-like effect by decreasing the immunocontent of proteins that constitute the NLRP3 inflammasome, including ASC, caspase-1, and thioredoxin interacting protein (TXNIP) in the hippocampus of Swiss male mice submitted to a stress model [319]. In another study, treatment of ovariectomized female Sprague Dawley rats with calcitriol (100 ng/kg, p.o., for 10 weeks) was shown to have an antidepressant-like effect, while inhibiting the NF-kB pathway and reducing the expression of iNOS, COX-2, and pro-inflammatory cytokines [320].

Vitamin D is a well-established immunomodulator [321,322]. Calcitriol (1,25(OH)2D3), the active form of this vitamin, is able to inhibit the adaptive immune system and positively modulate the innate immune system, thus improving the phagocytic activity of immune cells and decreasing the expression and release of inflammatory cytokines [323]. Particularly in the CNS, calcitriol is capable of inhibiting the activation of the NF-kB pathway, compromising the expression of iNOS, pro-inflammatory TLRs, components of the NLRP3 inflammasome, and pro-inflammatory cytokines, thus preventing microglial and astrocytic activation and neuroinflammation [324,325,326,327]. In addition to these mechanisms, calcitriol can also influence gut microbiota composition and promote the physical barrier established by gut endothelial cells by reinforcing intercellular junctions, thus reducing intestinal permeability [328,329,330].

## 9. Conclusions

In this review, we discussed how the dysregulation of the central and peripheral immune systems, as well as gut dysbiosis, can contribute to the pathophysiology of MDD (Figure 2). Preclinical and clinical studies have shown that factors such as stress, peripheral inflammation, and alterations in the gut microbiota may induce depressive symptoms. However, the mechanisms by which these factors initiate the neuroinflammatory process and induce MDD are not yet well understood. Therefore, it remains to be established whether specific immunological alterations can induce MDD, or whether different alterations in the immune system, which may be specific to each individual, contribute to the etiology of this disorder, making it difficult to establish a biomarker for MDD. Indeed, although it has been reported that individuals with depression have increased serum levels of inflammatory cytokines when compared to healthy individuals [10,204,331], the use of these cytokines as biomarkers for MDD presents various limitations, such as their low detection limit. Table 1 and Table 2 summarize several potential inflammatory biomarkers for MDD based on evidence from preclinical (Table 1) and clinical (Table 2) studies.

Nevertheless, several diseases that are related to a dysregulation of the immune system, including colitis, type-2 diabetes mellitus, autoimmune diseases, and COVID-19, have been repeatedly shown to be associated with an increased risk of developing MDD [3,224,332]. As such, modulation of the inflammatory process may have therapeutic benefits for patients with MDD. In agreement, several studies have demonstrated the anti-inflammatory effects of conventional antidepressants, as well as ketamine, although some discrepancies can be found in the literature with regard to the exact anti-inflammatory pathways related to the mechanisms of action of these antidepressants. As such, additional studies are still warranted so as to ascertain the exact therapeutic targets that can be modulated for the effective management of MDD. Finally, alternate therapeutic approaches, including physical exercise, probiotics, and nutraceuticals, have been shown to possess anti-inflammatory effects and to modulate gut microbiota, making them attractive for the treatment and management of MDD, particularly since inflammation has been associated with non-responsiveness to conventional antidepressant treatment [21]. As such, future research to explore their true therapeutic potential in MDD is a recognized priority.

## Figures and Tables

**Figure 1 cells-13-00423-f001:**
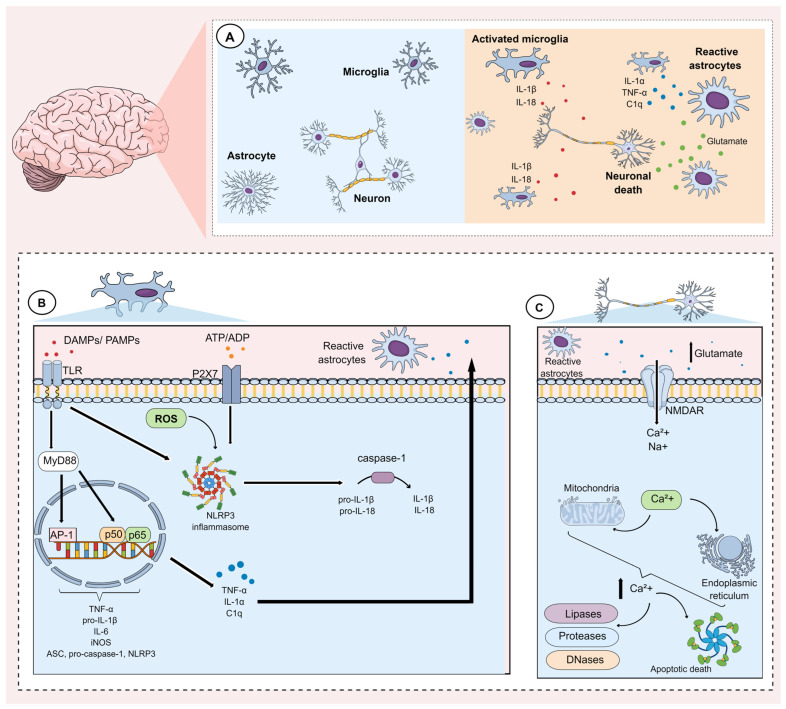
Mechanisms underlying microglial and astrocytic activation during neuroinflammation. (**A**) The neuroinflammatory process is mainly characterized by the activation of glial cells, such as astrocytes and microglia. Activation of these cells involves alterations in gene expression, which induce morphological and functional changes, characterized mainly by the synthesis and release of pro-inflammatory mediators. (**B**) Microglia activation can occur through pathogen-associated molecular patterns (PAMPs) as well as damage-associated molecular patterns (DAMPs). These factors activate Toll-like receptors (TLR) present on the microglial membrane. Once activated, these receptors activate the adaptor protein myeloid differentiation factor 88 (MyD88), which in turn modulates downstream pathways that promote the activation and translocation of nuclear factor kappa-B (subunits p50/p65) and the transcription factor activator protein-1 (AP-1). Together, these transcription factors induce the expression of tumor necrosis factor alpha (TNF-α), interleukin (IL)-6, pro-IL-1β, inducible nitric oxide synthase (iNOS), as well as components of the NLRP3 inflammasome (ASC, pro-caspase-1, NLRP3). Subsequently, the NLRP3 inflammasome can be activated via reactive oxygen species (ROS), P2X7 purinergic receptors, and TLRs that are activated by DAMPs and PAMPs, respectively. NLRP3 activation promotes caspase activation, which cleave pro-IL-1β and pro-IL-18 into IL-1β and IL-18, respectively. In addition, during microglial activation, mediators such as TNF-α, IL-1α, and complement component 1q (C1q) are synthesized and released. These mediators, in turn, activate astrocytes (reactive astrocytes). (**C**) Reactive astrocytes lose their ability to maintain glutamate homeostasis. They also tend to facilitate the release of glutamate. Consequently, a substantial increase in glutamate in the extracellular medium induces an intense influx of calcium ions (Ca^2+^) mainly via N-methyl D-aspartate (NMDA) receptors. This ionic influx favors an impairment in the ionic gradient of the mitochondrial membrane and the endoplasmic reticulum. The rupture of these organelles, in turn, induces the release of calcium from these intracellular stores. The resulting increase in intracellular calcium concentration results in the activation of enzymes that degrade proteins, lipids, and DNA. These mechanisms also favor apoptotic neuronal death.

**Figure 2 cells-13-00423-f002:**
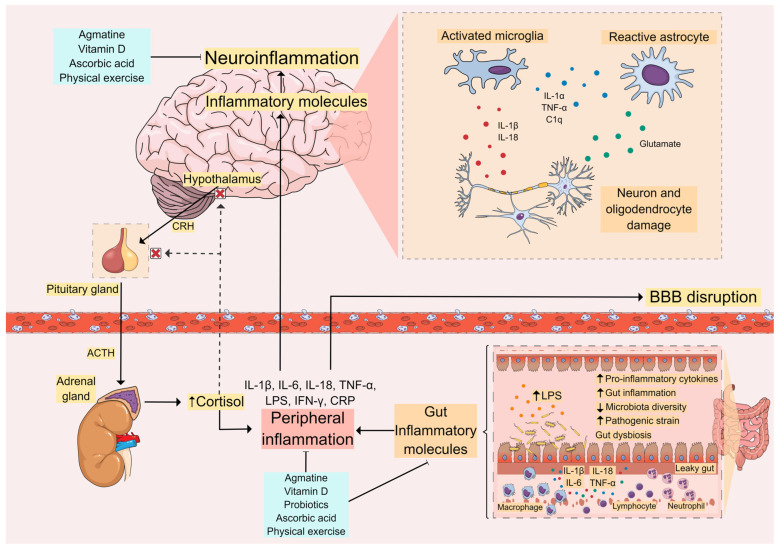
Inflammatory Pathways in MDD. Peripheral inflammation caused by stress, gut inflammation, and lipopolysaccharide (LPS) leakage activates the immune system and leads to increased levels of proinflammatory cytokines. This can cause blood–brain barrier (BBB) disruption, allowing these inflammatory molecules to reach the central nervous system (CNS). In the brain, these molecules contribute to neuroinflammation with the activation of glial cells and further release of proinflammatory cytokines, leading to activation of the hypothalamus-pituitary-adrenal (HPA) axis and cortisol release. This vicious cycle between peripheral inflammation and central neuroinflammation appears to be related to the onset of depressive symptoms and other comorbidities seen in MDD. Thus, molecules capable of controlling these inflammatory processes, such as agmatine, vitamin D, ascorbic acid (vitamin C), and probiotics, as well as non-pharmacological approaches such as physical exercise, are promising therapeutic strategies for the supplementary treatment and management of MDD.

**Table 1 cells-13-00423-t001:** Potential biomarkers of MDD: evidence from preclinical studies.

Biomarker	Animal Model	Sample	Studies
↑ IL-1β	Mice submitted to chronic stress	Hippocampus	[33]
		Hippocampus	[45]
		Serum	[238]
		Hippocampus	[262]
		Hippocampus	[265]
		Hippocampus	[296]
	Rats submitted to chronic stress	Cortex	[32]
		Prefrontal cortex	[236]
		Prefrontal cortex	[281]
	Mice submitted to chronic social defeat stress	Hippocampus and colon	[297]
	LPS in mice	Hippocampus	[45]
		Hippocampus	[231]
		Serum and hippocampus	[239]
		Hippocampus	[241]
	Fecal microbiota transplantation from inflammatory bowel disease patients, to mice	Blood and colon	[227]
	Ovariectomized mice	Hippocampus	[263]
	Rats submitted to an acute restraint stress	Serum, hippocampus, and prefrontal cortex	[284]
	Mice submitted to streptozotocin-induced type-II diabetes mellitus	Prefrontal cortex and hippocampus	[287]
↑ IL-18	Mice submitted to chronic stress	Hippocampus	[265]
	Ovariectomized mice	Hippocampus	[263]
	Rats submitted to an acute restraint stress	Serum, hippocampus, and prefrontal cortex	[284]
↑ IL-6	Mice submitted to chronic stress	Hippocampus	[34]
		Hippocampus	[33]
		Hippocampus	[45]
	Rats submitted to chronic stress	Hypothalamus and spleen	[32]
	Mice submitted to chronic social defeat stress	Hippocampus and colon	[237]
	LPS in mice	Hippocampus	[45]
		Hippocampus	[231]
		Serum and hippocampus	[239]
	Fecal microbiota transplantation from inflammatory bowel disease patients, to mice	Blood and colon	[227]
	Mice with inflammatory bowel disease induced by dextran sulfate sodium (DSS)	Rectum and hippocampus	[226]
	Rats submitted to high-fat diet and streptozotocin-induced type-II diabetes mellitus	Hippocampus	[286]
	Mice submitted to streptozotocin-induced type-II diabetes mellitus	Prefrontal cortex and hippocampus	[287]
	Rat model of type 2 diabetes mellitus and comorbid depression	Prefrontal cortex	[314]
↑ TNF-α	Mice submitted to chronic stress	Hippocampus	[33]
		Hippocampus	[45]
		Hippocampus	[278]
	Rats submitted to chronic stress	Hippocampus, cortex and spleen	[32]
	LPS in mice	Hippocampus	[45]
		Hippocampus	[222]
		Serum and hippocampus	[230]
	Mice with inflammatory bowel disease induced by dextran sulfate sodium (DSS)	Rectum and hippocampus	[217]
	Rats submitted to high-fat diet and streptozotocin-induced type-II diabetes mellitus	Hippocampus	[277]
	Mice submitted to streptozotocin-induced type-II diabetes mellitus	Prefrontal cortex and hippocampus	[287]
	Rat model of type 2 diabetes mellitus and comorbid depression	Prefrontal cortex	[314]

**Table 2 cells-13-00423-t002:** Potential biomarkers of MDD: evidence from clinical studies.

Biomarker	Study Design	Subjects	Sample	Studies
↑ IL-1β	Cross-sectional	Adults, 130 MDD and 40 HC	Blood	[28]
	Longitudinal	Young adults, 50 MDD treatment-naïve and 50 HC	Plasma	[183]
↑ IL-6	Longitudinal	Young adults, 50 MDD treatment-naïve and 50 HC	Plasma	[183]
	Longitudinal	Adolescents, 201 volunteered	Plasma	[182]
	Case-Control	Adolescent, 77 MDD and 54 HC	Serum	[180]
↑ CRP	Longitudinal	Young adults, 50 MDD treatment-naïve and 50 HC	Plasma	[183]
	Cross-sectional	Adults, 811 MDD	Serum	[181]
	Longitudinal	Adolescents, 201 volunteered	Plasma	[182]
↑ IFN-γ	Cross-sectional	Adults, 103 MDD and 97 HC	Blood	[179]

HC = healthy control; ↑ = increased level.

## Data Availability

Not applicable.

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
