# Peer review of "Role of Inflammatory Mechanisms in Major Depressive Disorder: From Etiology to Potential Pharmacological Targets"

_cells, 2024, doi:10.3390/cells13050423_

Round 1

Reviewer 1 Report

Comments and Suggestions for Authors

The review discusses the role of peripheral and central inflammation and gut dysbiosis in MDD pathophysiology and proposes alternative MDD therapeutic approaches. I have several comments:

- first of all, the relationship between inflammatory processes at the periphery as well as CNS and MDD has been previously discussed in several reviews and meta-analyses. Further, the MDD incidence with high rates of suicidality continuously increases in lower and lower age groups, mainly in adolescents. For these reasons, I recommend authors include more details on the differences at least in inflammatory factors as potential MDD therapeutic targets between MDD adults and adolescents

- similarly, the sex differences should be highlighted

- the authors also should include a paragraph highlighting the potential of the anti-inflammatory agents in MDD treatment

- the paragraph "Comorbidities associated with MDD" is not necessary in the context of the study

- line 15: please, include more commonly reported increased levels of pro-inflammatory cytokines in MDD such as IL-6 and TNF-alpha

- line 100: "neurotrophin" not necessary

- line 428: replace "reactivite" with "reactive"

- figure 2: please include in peripheral inflammation also IFN-gamma and CRP

-line 863: I recommend including " for the supplementary treatment"

Author Response

Reviewer 1

Point 1. First of all, the relationship between inflammatory processes at the periphery as well as CNS and MDD has been previously discussed in several reviews and meta-analyses. Further, the MDD incidence with high rates of suicidality continuously increases in lower and lower age groups, mainly in adolescents. For these reasons, I recommend authors include more details on the differences at least in inflammatory factors as potential MDD therapeutic targets between MDD adults and adolescents.

Response: Thank you for the suggestion. Please note that this information has been added to the sections “Role of the Peripheral Immune System in MDD” and “Antidepressant Effects of Anti-inflammatory Drugs”.

Point 2. Similarly, the sex differences should be highlighted.

Response: Thank you for the suggestion. Please note that this information has been added to the section “Role of the Peripheral Immune System in MDD”.

Point 3. The authors also should include a paragraph highlighting the potential of the anti-inflammatory agents in MDD treatment.

Response: Thank you for the suggestion. Please note that a specific section on this topic has now been added to the manuscript. This new section is titled “Antidepressant Effects of Anti-inflammatory Drugs”.

Point 4. The paragraph “Comorbidities associated with MDD” is not necessary in the context of the study.

Response: Thank you for the  suggestion. We have now removed this paragraph, as per the Reviewer’s suggestion.

Point 5. Line 15: please, include more commonly reported increased levels of pro-inflammatory cytokines in MDD such as IL-6 and TNF-alpha.

Response: Thank you for the suggestion. Please note that this suggestion was acknowledged and additional information regarding these cytokines have been added to the manuscript.

Point 6. Line 100: “neurotrophin” not necessary.

Response: Thank you for the suggestion. Please note that the term “neurotrophin” has been removed from this sentence.

Point 7. Line 428: replace “reactivite” with “reactive”.

Response: Thank you for pointing out this mistake, this has now been corrected in the revised version of the manuscript.

Point 8. Figure 2: please include in peripheral inflammation also IFN-gamma and CRP.

Response: Thank you for the suggestion. Please note that these markers have been added as requested.

Point 9. Line 863: I recommend including “for the supplementary treatment”.

Response: Thank you for the suggestion. Please note that this sentence was reworded as suggested.

Reviewer 2 Report

Comments and Suggestions for Authors

Although an interesting topic, there is lack of critical observation and opinions in the review. The information presented overlap with previous published review.

Norma Angélica Labra Ruiz, Daniel Santamaría Del Ángel, Norma Osnaya Brizuela, Armando Valenzuela Peraza, Hugo Juárez Olguín, Mónica Punzo Soto, David Calderón Guzmán, Inflammatory Process and Immune System in Major Depressive Disorder, International Journal of Neuropsychopharmacology, Volume 25, Issue 1, January 2022, Pages 46–53, https://doi.org/10.1093/ijnp/pyab072

Rahimian R, Belliveau C, Chen R and Mechawar N (2022) Microglial Inflammatory-Metabolic Pathways and Their Potential Therapeutic Implication in Major Depressive Disorder. Front. Psychiatry 13:871997. doi: 10.3389/fpsyt.2022.871997

Rahman, S., & Alzarea, S. (2019). Glial mechanisms underlying major depressive disorder: Potential therapeutic opportunities. Progress in molecular biology and translational science, 167, 159–178. https://doi.org/10.1016/bs.pmbts.2019.06.010

Afridi R and Suk K (2021) Neuroinflammatory Basis of Depression: Learning From Experimental Models. Front. Cell. Neurosci. 15:691067. doi: 10.3389/fncel.2021.691067

To restructure the abstract - lengthy introduction and objective are stated. The results and discussion are missing

Missing Introduction section - novelty, gap of the review, what are the main objectives of the review?

The findings can be summarised in a table form to provide a clear understanding eg Neuroinflammatory markers in preclinical models and clinical models of MDD

What are the limitation of the findings and future prospects/directions of the current knowledge?

The inflammatory biomarkers of MDD should be thoroughly explained.

Comments on the Quality of English Language

-

Author Response

Reviewer 2

Point 1. Although an interesting topic, there is lack of critical observation and opinions in the review. The information presented overlap with previous published review.

Response: Thank you for the comment. We would like to highlight that our revised version of the article provides a more integrative approach between the peripheral immune system, gut microbiota and neuroinflammation. In addition to describing the neuroinflammatory mechanisms that have  been addressed in more recent articles, we discuss how peripheral immune alterations, as well as alterations in the gut microbiota, can induce a  neuroinflammatory state that contributes to the onset and progression of MDD. In addition, we address different strategies that could be adjuvant to pharmacological treatment, particularly  nutraceuticals and physical exercise. This novel aspect has now received more emphasis in the revised version of our manuscript (abstract, introduction and conclusions).  Of note, throughout the text and in the conclusion we point out that future studies are needed to better understand the inflammatory mechanisms underlying MDD, which in turn will contribute to the identification of new therapeutic approaches for the treatment of this disorder, since current pharmacotherapy has important limitations. 

Point 2. To restructure the abstract - lengthy introduction and objective are stated. The results and discussion are missing.

Response: The abstract has been rewritten so as to address this suggestion.

Point 3. Missing Introduction section - novelty, gap of the review, what are the main objectives of the review?

Response: Thank you for this suggestion. Please note that the introduction has been restructured so as to emphasize the novelty and objectives of this review article.

Point 4. The findings can be summarised in a table form to provide a clear understanding eg

Neuroinflammatory markers in preclinical models and clinical models of MDD.

Response: Thank you for the  suggestion. We have now created two Tables summarizing the studies that investigated neuroinflammatory biomarkers in preclinical and clinical studies and these have been included in the revised manuscript.

Point 5. What are the limitations of the findings and future prospects/directions of the current knowledge?

Response: Please note that the conclusion was now been reframed, so as to emphasize areas that require further research and the importance of future studies in the identification of alternative therapies for the treatment of MDD.

Point 6. The inflammatory biomarkers of MDD should be thoroughly explained.

Response: Thank you for the  suggestion. Please note that inflammatory biomarkers in preclinical as well as clinical studies are described in three sections of the review, including "Neuroinflammation: Function of Glial Cells", "Role of the Peripheral Immune System in MDD", and "Gut Microbiota". Furthermore, and as above, we have now created two Tables summarizing the studies that investigated neuroinflammatory biomarkers in preclinical and clinical studies and these have been included in the revised manuscript.

Round 2

Reviewer 1 Report

Comments and Suggestions for Authors

The authors have addressed all comments.

I have one additional minor comment:- recently, the concept of a Compensatory Immune Response System (CIRS) has suggested the important role of anti-inflammatory markers in counter-regulating inflammatory effects in MDD. In this context, the part "Role of the Peripheral Immune System in MDD" could be further improved by including findings on pro-inflammatory/anti-inflammatory balance (ratios) in MDD adults, youth, and adolescents, also concerning the effect of sex.

Author Response

Reviewer 1

Point 1. Recently, the concept of a Compensatory Immune Response System (CIRS) has suggested the important role of anti-inflammatory markers in counter-regulating inflammatory effects in MDD. In this context, the part "Role of the Peripheral Immune System in MDD" could be further improved by including findings on pro-inflammatory/anti-inflammatory balance (ratios) in MDD adults, youth, and adolescents, also concerning the effect of sex.

Response: Thank you for the suggestion. We have added additional information to address this comment to the Section “Role of the Peripheral Immune System in MDD”. The new paragraph reads as follows:

"Overall, the evidence suggests that an activated inflammatory response system (IRS) contributes to the pathophysiology of MDD. This IRS is characterized by microglial, monocytic and lymphocytic activation, which culminate in the synthesis of inflammatory mediators, including TNF-α, IL-1β, IL-6, soluble IL-6 receptor (sIL-6R), IFN-γ, IL-2, and IL-17 [184,185]. However, some studies have also shown that some patients with depression exhibit increased Th2 and Treg activity, suggesting the presence of a compensatory immune response system (CIRS), which is characterized by increased levels of anti-inflammatory cytokines such as IL-4 and IL-10, as well as increased levels of soluble cytokine receptors (sIL-2R, sTNF-R1, sTNF-R2) and of the soluble IL-1 receptor antagonist (sIL-1RA) [185-188]. Indeed, some patients with depression have increased levels of both pro- and anti-inflammatory cytokines [188,189]. Therefore, components of CIRS probably counteract the effects of IRS in the context of MDD. The CIRS/IRS imbalance, in turn, could be a key factor in the development of a chronic inflammatory response observed in some MDD patients, particularly those who are resistant to treatment with conventional antidepressants [184]. In view of this, several studies have assessed the levels of these mediators, so as to better understand the relationship between CIRS/IRS in MDD. For example, studies included in a meta-analysis found an association between increased levels of IL-1RA, IL-6, IL-10, IL-12, sIL-2R, sIL-6R and TNF-α and decreased levels of IFN-γ and IL-4 in adult patients with depression [190]. However, the levels of these markers seem to vary between studies, especially considering variables such as age and sex. In a recent study conducted with adolescents with depression, it was possible to see a significant increase in some markers such as IL-4 and Treg+Th2, while these have not been observed in most studies conducted with adults. Furthermore, when the analyses considered potential sex differences, it was found that female adolescents with MDD only had increased levels of IL-10 and TNF-α, while male adolescents with MDD had increased levels of IL-4, IL-10, sIL-6R, Treg+Th2 and TNF-α/TNF-R1 [191]. Finally, a study conducted by Sowa-Kućma et al. [192] reported that the severity of MDD, measured with the Hamilton Depression Rating Scale, was correlated with an increase in IRS and CIRS markers, including sIL-6R, tumor necrosis factor receptor 80kDa (sTNFR80), and zCytR (z-unit weighted indices reflecting the 5 cytokine receptor levels). This study also showed that previous suicide attempts are associated with increased sIL-1RA, and IL-1α levels [192]".